# Benchmarking Giant Panda Welfare in Tourism: A Co-Design Approach for Animals, Tourists, Managers, and Researchers

**DOI:** 10.3390/ani14152137

**Published:** 2024-07-23

**Authors:** Yulei Guo, David Fennell

**Affiliations:** 1Tourism Department, Chengdu Research Base of Giant Panda Breeding, Chengdu 610081, China; 2Biology Department, Oulu University, 90570 Oulu, Finland; 3Geography and Tourism Department, Brock University, St. Catharines, ON L2S 3A1, Canada; dfennell@brocku.ca

**Keywords:** giant panda (*Ailuropoda melanoleuca*), co-design, benchmark, animal welfare, tourism, visitor comments

## Abstract

**Simple Summary:**

This study presents a new framework for understanding how tourists perceive animal welfare, incorporating insights from tourists, researchers, and the animals themselves. Using scientific theories to define key aspects of animal welfare, the framework is refined based on tourist feedback, making it a reliable tool for assessing these perceptions. At the Chengdu Research Base of Giant Panda Breeding, we analyzed 4839 visitor comments from March to August 2023 to evaluate perceptions of panda welfare. This method highlights the importance of clear communication in educational programs, aiming to boost public understanding of animal welfare. By addressing the diverse and complex views of tourists, the framework supports more effective conservation education. It shows that collaborative efforts can create a flexible and theoretically sound benchmarking system, enhancing tourist education and fostering a stronger commitment to animal welfare and conservation as essential elements of sustainable tourism.

**Abstract:**

This study introduces a co-design benchmarking framework to understand tourists’ perceptions of animal welfare, integrating diverse perspectives from tourists, researchers, and animals. By leveraging scientific theories to establish benchmark dimensions, the framework is refined through visitor input, ensuring a robust and adaptable methodological tool for assessing tourists’ perceptions and animal informed consent in wildlife tourism. Using the Chengdu Research Base of Giant Panda Breeding as an example, we analyzed 4839 visitor comments collected from March to August 2023 to benchmark perceptions of giant panda welfare. This approach underscores the importance of effective communication in educational initiatives, aiming to enhance public literacy and knowledge about animal welfare. By addressing the complexity and variability in tourists’ perceptions, the proposed framework contributes to more impactful conservation education efforts. The study demonstrates that a collaborative effort results in a benchmarking framework that is firmly grounded in theoretical foundations yet flexible enough to adapt based on visitors’ insights and animal participation. Ultimately, this comprehensive approach ensures that educational initiatives resonate with tourists’ diverse backgrounds, fostering a deeper understanding and commitment to animal welfare and conservation, which, we argue, should be key components of sustainable tourism.

## 1. Introduction

Animal care organizations and wildlife alliances around the world are prioritizing their responsibility to educate tourists on animal welfare and conservation. Wild Welfare [1] states that “wildlife tourism is a huge market and can promote local livelihoods, education, and conservation”. For the Association of Zoos and Aquariums [2], conservation education is vital to the organization’s government affairs strategy because visitors can experience a stronger connection to nature in zoos and aquariums. Likewise, researchers have reached the consensus that education is urgently needed to improve the public’s understanding and knowledge of animal welfare and conservation [3,4,5,6,7,8,9]. However, educating the public in the context of wildlife tourism may face numerous challenges for visitors [6,10,11,12], even though the quality and quantity of care for animals should be intrinsic to the vision of sustainable tourism [13,14,15].

One suggested reason for this gap is that public perceptions of animal welfare often depend on many biological and sociocultural factors [16]. For example, drawing on tourist boycotts and protests on animal abuse, Shaheer, Carr, and Inch [17] found that tourists’ participation depends on several factors, including awareness levels, local political dynamics, ideological beliefs, accessibility, and the severity of the animal welfare issue prompting the boycott. Miele et al. [18] suggest that public perception of animal welfare can diverge from a unified definition because of its link to broader differences in ethical and value-based perspectives on human–animal relationships. Studies also indicate that the public’s perception of animal welfare often leans heavily toward emotions, which can result in judgments based on intuition [19,20,21]. Hölder, von Meyer-Höfer, and Spiller [22] point out that laypeople’s agreement or rejection of animal-ethical intuitions can be species-specific depending on classifications such as ‘edible’ or ‘inedible’. The challenge of accurately assessing and understanding animal welfare using scientific measures and knowledge also leads to the tourists’ animal welfare illiteracy [6]. 

Although there is growing interest in animal welfare in tourism, this interest is not the major driver of tourist-consuming decisions because perceptions of animal welfare have played an indefinite role in the growth of the industry in comparison to the price, quality of animal encounters, and pleasure of the experience [23]. Carr and Cohen [11] suggest that despite contemporary zoos having often listed conservation and education in their mission statement, zoos still prioritize entertainment when communicating with visitors. The focus on profit and pleasure blinds people from understanding indicators of whether animals assent or dissent to participation as tourist attractions and enablers. While recognizing the importance of comprehending people’s views on animal welfare to enhance awareness, change attitudes, and improve communication between animal venues and the public, the complex and emotional nature of public perception and prioritization of profit and pleasure presents significant obstacles for conducting targeted studies in this area. 

In an effort to more formally move away from the dominant anthropocentric and contractarian ethos in animal-based tourism, this study uses a co-design benchmarking framework to understand tourists’ perceptions of animal-informed consent, while also introducing more care, empathy, and understanding towards the use of animals in wildlife tourism. Benchmarking is a well-established practice in fields including corporate ethics [24,25], medical ethics [26], and academic integrity [27] to measure and improve an organization’s ethical performance. In farm and lab animal research, where animal welfare has been a more widely studied and historically established topic, benchmarking has recently been introduced as an effective tool to enable cross-cultural comparisons [28,29,30,31], measure the actual outcomes of welfare [29,30,32,33,34], and demonstrate accountability [35,36]. Co-design, or collaborative design, is an attitude and set of methods and tools that encourages new ways to work with others in an “‘other-regarding’ ethics” (p. 2298) that seeks to build empathy, stewardship, and care in efforts to be more sustainable in tourism [37]. 

We amplify this focus on others in co-design by including the interests of animals in tourism enterprises and experiences in our benchmarking tool to encompass a diverse range of perspectives. As researchers, we leverage scientific theories and frameworks to establish benchmark dimensions and indicators, which are further expanded and refined based on input from visitors and animals. Our approach demonstrates that this collaborative effort results in a benchmarking framework firmly grounded in theoretical foundations yet flexible enough to adapt and specify animal welfare considerations based on these visitor and animal insights and reflections. This framework thus provides a robust methodological approach for assessing and responding to tourists’ perceptions of animal welfare and animals’ participation within the context of wildlife tourism. Benchmarking is used as a foundation for ensuring that educational initiatives are communicated effectively and resonate with tourists’ diverse backgrounds, ultimately contributing to more meaningful and impactful efforts to enhance animal welfare literacy and knowledge among the public. 

The case study site of our research is the Panda Base. Building on the giant panda informed consent quantitative evaluation developed by Fennell and Guo [38], the present paper builds on this previous work through a thematic and sentiment analysis of 4839 visitor comments collected from Panda Base from March to August 2023. While questionnaire data could suffer from issues such as response bias, question design, and social desirability bias [39,40], visitor comments, as voluntary outlets, are often considered “authentic, trustworthy, and helpful” [41] (p. 608). Hence, visitor comments and their focus provide insight into the validity and feasibility of benchmarking giant panda welfare based on an animal-informed consent framework. In this study, we first investigate how benchmarking has been used to improve farm animal welfare and how wildlife tourism could benefit animals by incorporating the benchmarking method. We then review giant panda informed consent evaluation and its rationale, before presenting the design of our study.

## 2. Literature Review

### 2.1. Benchmarking in Animal Welfare 

In the 1960s, individuals and organizations in Western Europe amplified concern over farm animal welfare, eventually spreading globally with a focus on establishing national animal welfare legislation [42]. However, the emphasis shifted within Western Europe towards a unified cross-national legal framework under the European Union (EU). Despite this common EU legislation, its implementation varied across European countries. Some nations adhered to minimum standards, while others sought to surpass them [43]. International competition prompted efforts to achieve global agreements on higher animal welfare standards, but progress was challenging. Consequently, a market-driven approach to animal welfare emerged, where products meeting stricter welfare standards were labelled and sold at premium prices, benefiting both farmers and consumers. Combining legislation-led and market-driven standards, this dual approach met consumer demand in complementary ways: state-led regulations provided support and structure, while market-driven initiatives offered flexibility and variety [31].

Despite advancements, measuring improvement in animal welfare remains challenging, especially between nations. Efforts have been made to facilitate cross-national comparisons, including the Business Benchmark on Farm Animal Welfare (BBFAW, [44]) and the Animal Protection Index (API, [45]). BBFAW evaluates food companies based on policy commitments, governance, performance impact, and more, encouraging transparency and best practices. API assesses national animal welfare policies and legislation across various categories, using a seven-point scale to rate countries. Another approach involves country-based assessments focusing on representative farms, exemplified by the EU-funded Welfare Quality project. Sandøe et al. [30] introduced a benchmarking approach for cross-country comparisons in pig production, emphasizing specific parameters valued by experts. This method tailors assessment criteria based on project objectives, adopting a transparent numerical system for calculation and weighting. Such initiatives contribute to improving animal welfare standards while addressing the complexities of cross-national comparisons and benchmarking.

### 2.2. Benchmarking in Tourism

Benchmarking involves a systematic comparative measurement for continuous improvement [46]. Widely used in management practices, benchmarking is particularly impactful when applied to processes that, when optimized, significantly enhance a business’s competitiveness and performance [46,47,48]. Research suggests that benchmarking is more powerful than mere competitive analysis as it optimizes company procedures and ensures customer satisfaction [49,50,51].

While the hospitality sector has seen numerous benchmarking initiatives, tourism destination benchmarking has gained attention, especially in sustainable tourism [49,50,51]. For instance, Font et al. [52] used benchmarking as a comparative between destinations as part of a list of knowledge transformation challenges that introduce sustainability measures based on evidence from indicators. Cernat and Gourdon [53] proposed a benchmarking tool to assess sustainability weaknesses in tourism destinations and inform policymakers on improvements. The tool encompasses seven dimensions, including assets, activity, linkages, and sustainability, further broken down into variables and key indicators. Blancas et al. [54] introduced a dynamic, sustainable tourism evaluation tool using a composite indicator, reflecting a destination’s progress toward sustainability goals. Li and Wu [54] used tourist surveys to assess pro-environmental behavior and understand tourists’ behaviors based on eight dimensions, highlighting the flexibility and utility of benchmarking in research and industry.

Tourism scholars and industry have used benchmarking to evaluate a business’s progress or to enhance tourists’ understanding of a phenomenon. Benchmarking can be a flexible and useful tool whose structure, aim, and methods can be modified to meet a project’s specific aim. Additionally, cross-disciplinary approaches have been used to structure benchmarks, where the construction of a tree of indicators is often the first key step. In this study, we use Fennell’s [55] animal-informed consent framework, adapted and used by Fennell and Guo [38] in the context of giant panda tourism, as the dimension-indicator framework on which our benchmarking procedure is constructed.

### 2.3. Giant Panda Welfare and Informed Consent

Animal-informed consent builds on the Five Domains model of animal welfare, which has been a widely recognized framework for evaluating the well-being of animals in various contexts [56,57]. The Five Domains model encompasses (1) nutritional conditions and their impacts on achieving an optimal body condition score; (2) physical environmental conditions and their effects, such as access to fresh air, good ventilation, and shelter from adverse weather; (3) health conditions and their effects, encompassing aspects like hygiene, disease management, injuries, nutrition, and access to veterinary care; and (4) behavioral interactions—both with the environment and with other animals, as well as human interactions. The last domain, the animal’s overall mental state and subjective experience, is evaluated by integrating these four domains. In their most recent development of the Five Domains model, Mellow et al. [58] suggest that the model needs to be extended to facilitate explicit and detailed assessment of impacts caused by humans, such as livestock handlers, zookeepers, wildlife managers, and researchers. 

Recognizing the impact that the tourism industry has on animals used for entertainment, Fennell [55] argued that animals themselves indicate physiological and psychological cues that would allow handlers to understand the degree to which animals choose to participate in tourism events and episodes, if at all. Three main consent states identified by Fennell, including active (extremely positive and moderately positive affective impact), passive (negligibly positive and negligibly negative affective impact), and no animal-informed consent (moderately negative and severely negative affective impacts) based on indicators established by the Five Domains model. Using a case study of sled dog tourism, Fennell [55] shows that the animal-informed consent framework can help tourism stakeholders place the interests and preferences of sled dogs on a higher plane. 

Fennell and Guo [38] developed an initial application of animal-informed consent at the Chengdu Research Base of Giant Panda Breeding—one of China’s most visited wildlife tourism attractions (for more details and a map of Panda Base, please see Fennell and Guo [59]). Under the domain of behavioral interactions, the giant panda’s interaction with tourists was measured based on two indicators, namely, the giant pandas’ relaxed presence in front of the tourists and their responses to tourists as a function of how tourists interacted differently with the giant panda as compared to the conceptual model of sled dogs used by Fennell [55]. The empirical context in which the framework was adapted meant the researchers [38] surveyed tourists. Similar to the benchmark of Li and Wu [60], this tourist-focused approach suggested using layman-friendly language when transforming the welfare framework into a tourist questionnaire. A 6-point Likert scale format was used to articulate the animal-informed continuum. 

This study builds on the giant panda informed consent as a framework to evaluate tourists’ understanding and perception of giant panda welfare. According to Fennell [55], empirical data are needed to support the development of animal-informed consent indicators. By expanding the giant panda informed consent into a benchmark, this study investigates how the tool can reflect on visitors’ perceptions of giant panda welfare. Furthermore, it foregrounds a baseline from which to develop future work. 

## 3. Methodology

### 3.1. Research Background: Giant Panda Welfare in Tourism

The welfare of giant pandas in zoos has historically lacked consistent and systematic attention despite their status as one of the planet’s most beloved animals. Early attempts to hold giant pandas in modern zoos were more about institutional profit and visitor pleasure than the welfare and conservation of these animals. Limited scientific knowledge, for example, kept zoos feeding panda cubs with porridge supplemented with eggs, fruit, and specific vitamins until the mid-1990s, resulting in the animals experiencing vomiting, lethargy, and diluted feces [61]. Schaller [62] noted that the panda’s welfare had been utterly disregarded by zoos and other international organizations, including WWF, IUCN, AAZPA, and the International Union of Directors of Zoological Gardens, to the point where a worldwide temporary moratorium had to be implemented on panda loans [61].

Following Armand David’s initial encounter with the giant panda, observing the giant panda and successful breeding became the priority and foremost challenge for animal organizations hosting giant pandas [61,62,63]. According to Nicholls [61], because of the consistent challenges zoos faced in successfully breeding giant pandas, zoos started to take more care of the panda’s enclosures as impoverished captive environments contributed to unsuccessful breeding attempts. Environmental enrichment plans started to take shape in giant panda enclosures following the animal husbandry principle that addressed animals’ psychological and physiological well-being [61]. However, even at this stage, the welfare of the giant panda remained secondary to the main priority of giant panda breeding.

The construction of spacious and well-designed enclosures attracted tourists, whose entertainment demands soon elevated alongside the scientific exploration of giant panda breeding. Zoos strategically marketed the giant panda to elevate visitation and enhance visitor satisfaction. For example, in 2004, Shanghai Zoo initiated the ‘hold a panda’ experience for visitors [64], which launched a lengthy debate between the government, public, and animal organizations regarding panda welfare and visitor needs. It was not until the giant panda’s health was seriously jeopardized by zoonotic transmission between visitors and pandas that the Chinese authority finally ended the practice in 2018. At the same time, volunteer programs claiming opportunities to make intimate contact with the giant panda prospered in both Chinese and international zoos. Although scholars [65,66] believe that giant panda conservation has been successful, we note in this study that giant panda conservation does not necessarily mean an improvement in giant panda welfare, especially for giant pandas in captivity. Giant panda welfare is secondary to panda breeding programs and visitor demands at Chinese and international zoos.

The return of giant panda Ya Ya from the Memphis Zoo to China in April 2023 was welcomed by Chinese social media and animal welfare activists online. A video of Ya Ya in Beijing posted by Chinese state media CCTV received more than 200,000 likes within one day as an expression of national pride [67]. While the notion of ‘panda diplomacy’ [68,69] has captured the political implications underlying China-U.S. relations, social media, panda fans, and animal activists have played indispensable roles in the online campaign against the poor treatment of pandas. According to Li [70], animal lovers, pet owners, and their supporters have become an increasing force in China “whose voice is heard more and more often” (p. 312). The result is that the giant panda, the ‘new cat’ of the Internet [71], has become a ubiquitous Internet celebrity animal. The intense affection and emotion between the giant panda and the public fan community [72], exemplified by the case of Ya Ya, suggests that the welfare of the giant panda, a task ignored historically by animal organizations, is now surfacing as a primary public concern. A co-designed benchmark will assist the transition into a more caring society towards animals while providing possible baselines and guidance.

### 3.2. Co-Design Animal Welfare in Tourism

Given the public’s growing interest in giant panda welfare, and the impactful nature of tourists’ online reviews as an asset for decision-making [73], we believe it is critical to benchmark giant panda welfare both through and with the public perspective. Unlike information generated by third parties or business organizations, tourist reviews are often deemed credible, authentic, and reliable sources of information [74,75,76,77]. Currently, tourists in online review communities are considered active co-designing participants in generating value and a source of marketing information with tourism venues [78,79,80]. This paradigm shift empowers the tourist community, pointing to directions of knowledge creation, innovation, and marketing opportunities for giant panda welfare in tourism. 

Liburd, Deudahl, and Heape [37] suggest that co-design is a central tenet for intentional change in sustainable tourism. Smit et al. [81] call for conscious and reflexive stakeholder participation in tourism activities and decision making. According to Smit et al. [8] (p. 8), co-design involves intentionally collaborating with a diverse group of stakeholders to explore and address complex problems. This process leverages collective creativity and generative design thinking to uncover various perspectives and develop effective solutions. Coghlan [82] enlisted co-design through an Action Design Research approach in the context of the Great Barrier Reef to bridge the gap between practitioners and researchers. Such included marine biologists, reef tourism operators, reef educators, park managers, and tourists. 

Thus, co-design provides the rationale behind the design of this study in which, on one hand, the benchmark aims to integrate tourist reviews to shed light on the current knowledge of animal welfare among this stakeholder group. On the other hand, Fennell’s [55] animal informed consent framework elevates the status of animals as critical participants contributing value to tourism experiences. In this light, our co-design benchmark encourages the voices of animals, tourists, and tourism researchers to innovatively shape this touristic space. First, the giant panda-informed consent is used as the framework for benchmark dimensions and indicators. Second, available data sources are studied and identified. Third, the analytical method is developed based on the type and nature of the data sources. Fourth, study results are presented, and these results are discussed in the context of future directions.

### 3.3. Data Collection and Analysis

On 24 March 2023, Panda Base introduced an online visitor book (OVB) for tourists on its official WeChat account in response to the growing volume of visitors after the pandemic. The OVB was listed as a sub-banner under the ‘Tourism Service’ window, where tourists could access an electronic map, purchase tickets, and consult the service hotline. The OVB allows visitors to write suggestions and comments about their experiences at the Panda Base. The OVB is built on Wenjunxing, a professional website that provides access to questionnaires and other forms of information. Tourists using the OVB can choose to offer their demographic information, such as gender and mobile phone number, but are mandated to identify a category for their comments, which includes Ticketing, Sightseeing bus, Infrastructure, Employee attitude, Panda viewing experience, and Others. 

As of 24 August 2023, 4839 comments were collected over 154 days on the OVB site, meaning that, on average, 31 comments were written into the OVB daily. In total, 11.9 million tourists visited Panda Base in 2023 [83], averaging about 33,000 visitors per day. Eighty repetitive and three invalid comments with only punctuation marks were deleted from the dataset, leaving 4756 valid entries. Unlike a traditional visitor book often found at the Visitor Center or Reception Desk, the OVB is a one-way communication device between visitors and the Panda Base, with visitors identifying issues and providing possible management reactions. Only management at Panda Base can access each comment, and visitors have no further control over how their comments are processed. Most submitted comments were written in Chinese, with less than ten entries in English.

Based on the paper’s aim, 1246 (25.75%) comments were marked under the ‘giant panda welfare’ category, where tourists expressed their concerns for pandas at the Base. How visitors think for and about the giant panda, rather than the visitor experience, was considered the thematic threshold for giant panda welfare. The coding process took approximately a month for the researcher and her researcher assistant, with initial coding by the researcher after careful instruction. The researcher randomly verified 10% of the coded data in NVivo 14. If a disagreement was found, the researcher discussed it with the research assistant to identify the right place for the comment. This process continued until consistency was obtained between the two researchers. After these iterative checks, the full dataset was coded. 

To better understand the comments and how tourists were addressing the welfare of giant pandas, the coded giant panda welfare comments were further categorized (Table 1). We note that a comment may address several aspects of the giant panda welfare indicators. Hence, comments are coded multiple times under different indicators (see Figure 1 for an example). The example in Figure 1 was identified as positive based on Figure 2, which shows that visitor comments were labelled into negative, neutral, and positive categories as a function of sentiment. The research team categorized the sentiment of each comment after coding the comment for its specific welfare indicator(s).

## 4. Results

Table 2 presents a co-designed version of the giant panda welfare informed consent indicators and how this version is different from the framework [38] proposed. Table 2 shows that the research-based framework has provided a solid ground to which visitors collectively responded. However, indicators are both added and excluded from the co-designed framework, showing that tourists were able to tend to specific welfare issues about the giant panda (e.g., the addition of Indicators 2d, 2e, 3d, 3e, 4.2d, 4.2e, 4.4c, 4.4d, 4.4e and 4.4f). In contrast, the research-based framework addressed the researchers’ concerns (Indicators 4.4a and 4.4b) more than the tourists. Table 2 shows that tourists can enrich and consolidate giant panda welfare indicators through their practical interactions with the animal. Nevertheless, the research-based framework is needed to provide scientific categories to benchmark the visitor comments that are otherwise discrete and scattered. Indicators 4.4a and 4.4b in giant panda informed consent, aimed at evaluating if pandas assented or dissented to being used as tourism attractions based on the perception of visitors. While the survey conducted in Fennell and Guo [38] mandated participants to reflect on the two indicators, Table 2 shows that visitor comments did not specifically address whether pandas were active agents offering consent to being used as tourist attractions. Furthermore, how pandas responded to visitors’ attention was not of concern for respondents. 

Figure 3 shows the frequency of visitor comments and the distribution of three emotions (negative, neutral, and positive) in the co-design giant panda welfare framework. Figure 3 shows a highly uneven frequency distribution in the visitor comments. The top three concerns of tourists were 4.1c, Multiple choices to move and be active; 3a, Suffering from acute or chronic injuries; and 2c, Enclosures that protect pandas against thermal extremes, all having more than 200 reviews. We can see from Figure 4 that most tourists tended to express their ideas negatively or positively rather than neutrally. Also, the distribution of emotions in each indicator can vary based on descriptions included. This means that no indicator has been completely positive nor negative.

The bubble chart of Figure 4 also demonstrates visitors’ emotional intensity towards each indicator (visible also from the zero line) and the frequency of visitor comments in each indicator (visible also from the size of the bubble). It is evident in Figure 4 that 4.1c, 3a, and 2c are the top three indicators addressed by visitors, with more negative than positive comments in 4.1c. Figure 4 shows that Panda Base needs to tackle 2e (Quiet environment), 4.4e (Tourists follow rules), and 4.2d (Giant panda receive equal treatment) as the priority in order to improve visitors’ understanding of the giant panda welfare since all three were negatively perceived at a high frequency (e.g., with more than 100 comments). Notably, all these are co-designed indicators that visitors added into the giant panda welfare indicators.

The bubble chart of Figure 5 demonstrates the emotional intensity visitors held towards each indicator and compares this result with the previous survey. Similarly, the bubble size shows the frequency of the visitor’s comments. The study by Fennell and Guo [38] demonstrates that visitors at Panda Base were satisfied with all giant panda indicators (Table 1), with all indicators reaching more than 5 points (based on the 6-point Likert scale). Indicator 3b (Healthcare treatment) shows that pandas received good healthcare at the Panda Base, with visitor comments and survey results positively addressing this indicator. In contrast, visitors were more likely to address Dimension 1 (nutrition) most negatively despite these having been considered excellent in the survey. Indicator 4.1c (Multiple choices to move and be active) was mostly negatively addressed, but we also note its mediocre performance in the survey. In contrast, 3a (The giant panda does not suffer from acute or chronic injuries) was more positively depicted in visitor comments than in the survey. Notably, visitor comments and surveys negatively depicted include 5a, 5b, 5c, and 2a. 

Of the 1251 comments categorized as ‘giant panda welfare’, 81.6% (*n* = 1021) of the comments were written for particular panda celebrities with the panda’s name included (e.g., Huahua, Runyue, and Yuanrun). Two-hundred and thirty (18.4%) comments addressed giant pandas as a species or in general. Text coverage shows that comments addressing celebrity pandas produced more than four times the comments than those written for pandas. Fennell and Guo [38] addressed the impact of giant panda fandom on visitors’ perception of giant panda welfare. The significant impact of giant panda celebrities and the fan culture again surfaced in visitor comments. Figure 6 provides a closer view of how the celebrity and regular pandas were addressed in visitor comments. 

## 5. Discussion

Benchmarking has been used in industrial practices as a strategic management tool and process to compare and measure an organization’s performances, practices, and outcomes. Benchmarking also provides tools for comparison between similar organizations and industries. In this study, we show that both animal welfare and the tourism and hospitality industry have implemented benchmarks to achieve improvement and identify opportunities to enhance performance and competitiveness. The recent development in wildlife tourism presents an urgent need to evaluate and ensure animal welfare in tourism practices as well as the necessity to improve tourist satisfaction and animal welfare literacy [6,84,85]. This paper developed a co-design animal welfare benchmarking tool based on expert knowledge, visitor surveys, and comments of the welfare conditions of giant pandas at the Base. The benchmark aims to offer an overview of current animal welfare evaluated and perceived by tourists based on the theoretical framework given by tourism scholars. This co-design framework also provides a starting point for tourism scholars and organizations interested in improving conservation programs’ effectiveness and tourists’ animal welfare literacy. The benchmark should allow further collaboration between tourism scholars, wildlife organizations, and tourists on theoretical and practical levels. 

The need to construct a giant panda welfare indicator framework originated from one of the researcher’s management tasks at the Panda Base. The sheer volume of visitor comments and their discrete nature suggested that a systematic approach would be most helpful in organizing and realizing the value of these comments for the management at Panda Base. This managerial need meant a framework in which different comments could be categorized and tagged. The researcher’s familiarity with her own research work provided the first clue to work on the tourist comments. The giant panda informed consent indicators used in Fennell and Guo [38] have already addressed essential aspects of giant panda welfare. While the researcher categorized and identified the visitors’ comments under each category, it emerged that tourists also engaged in aspects of giant panda welfare untouched in the previous theoretical-based framework. For instance, neither the Five Domain framework nor the giant panda informed consent had specifically addressed issues of 2d. humidity, and 2e. noise at the giant panda enclosures, yet tourists’ comments raised these two issues. Indeed, studies have demonstrated that both humidity [86,87,88] and noise [89,90,91] can impact animal welfare, especially captive animals. Similarly, tourists also demonstrated concern towards 3d stereotypic behaviors [92,93] and 3e breeding [94,95], which are also established research topics in animal welfare. 

4.2d Giant pandas receive equal treatment (137 comments), we believe, is one of the unique indicators that visitors added to the giant panda welfare benchmark and is specifically relevant to giant panda tourism on a contextual level. Fennell and Guo [38] mentioned that giant panda fandom could be a crucial factor in differentiating the fans’ interest in giant panda welfare from non-fans. We note that the phenomenon of giant panda celebrity [72] has been well-identified by visitors who also suggested the relationship between animal celebrity and animal welfare. For example, comment No.195 made the request: 

“Treat every giant panda well! Whether celebrity or not, all pandas should be treated equally! Avoid overbreeding! Say no to scientific experiments!”

This celebrity-regular gap in pandas can be contextualized locally to Panda Base due to the large number of giant pandas on display. The pursuit of equality for pandas as an important aspect of giant panda welfare, we believe, will add to our existing understanding on animal celebrity [96,97] and the different appeals between charismatic and non-charismatic species [98,99,100]. As the benchmark shows, it is fair to suggest that the current concern towards giant panda welfare is panda-celebrity-driven. We suggest further studies are urgently needed to understand how this focus on the panda celebrity might influence visitors’ overall understanding of giant panda welfare and whether the giant panda celebrity, as a phenomenon, could exert a more positive influence on their welfare. 

Indicator 4.2e ‘Companion animals’ (6 comments) draws on companion species and parallels the giant panda with pets. Companion animals were mentioned in opposite contexts. Either the giant panda was likened to pets (e.g., demanding the same emotional investment and care for giant panda) or opposed to pets (e.g., differing giant panda from the pets because the giant panda needs ‘respect’ more than interest). Indicators 4.2e and 4.2d enriched the dimension of the giant panda’s interaction with other animals. We note that the research-based framework did not address these two indicators. 

Indicator 4.4. ‘Tourists’ was the dimension tourists most actively engaged with. Notably, tourists edited out the 4.4a and 4.4b proposed by researchers, while four more indicators, 4.4c, 4.4d, 4.4e, and 4.4f, were added. Indicator 4.4c Information accessibility targeted communication between the Panda Base, the fourth most important aspect of the Panda Base. Researchers [18,101,102] have long called for establishing a dialogue between science and society on animal welfare issues. It becomes obvious to us that tourists at Panda Base are building a stronger awareness of giant panda welfare issues, which demands an effective and more strategic response from the organization. Indicator 4.4b Tourists empathize with pandas, refers to emotions and feelings tourists are willing to invest in animals. Studies demonstrate a strong link between public empathy and support for animal welfare [103,104,105]. Indicator 4.4d Tourists follow the rules, taps into the ongoing discussion on tourists’ self-regulation in tourism [106,107,108]. These studies have specifically drawn on sustainability in tourism, which can be relevant to tourists’ self-regulation in giant panda welfare. Indicator 4.4d Panda-related activities, expand the scope of other related giant panda welfare issues, such as birthday parties for pandas. 

The co-design framework shows that visitors and experts in animal welfare can collaborate to expand and broaden the welfare landscape of a species such as the giant panda. Using the Five Domains framework and its adaption in the giant panda informed consent indicators laid the foundation for addressing welfare issues in giant panda tourism. Notably, visitor comments had themes well-developed under each indicator. However, we believe this co-design framework showcases the value of tourists’ participation and insight into animal welfare in tourism. The inclusion of visitor comments enabled the researchers to expand the benchmark and tailor the framework to the specific context of giant panda tourism at Panda Base. Based on existing research, the co-design framework added welfare indicators that can be built into the benchmark. This benchmark tool also demonstrates its value on the managerial and practical level for its ability to inform management of the ongoing change and visitors’ experiences that are otherwise highly discrete and inconsistent, such as individual comments. Mapping the visitor comments based on the established giant panda consent indicators allows both the questionnaire survey and the visitor comments to merge and offers a different perspective of visitors’ interest and emotional intensity on different indicators. Future studies should investigate whether operators are able to assess, specifically, the degree to which pandas assent or dissent to being used as tourism attractions as proposed by Fennell [54] according to the Five Domains indicators developed by Mellor et al. [58]. 

The co-design benchmark shows that the giant panda’s use as a tourist attraction is a taken-for-granted reality that visitors do not consciously reflect on in their comments. Although we read comments suggesting that the pandas were happy and having a good time, these comments had not explicated the presence of tourists as factors contributing to pandas’ consent to being used as tourist attractions. We note questionnaires can play a critical and complementary role to the collection of reviews where the perceptions and opinions representing the interests of giant pandas can be specifically addressed and collected. 

This benchmark tool allows the integration of qualitative and quantitative data and provides flexibility in expanding its framework as co-design proceeds. Furthermore, the benchmark enables longitudinal studies that allow researchers and management to expand and modify welfare indicators over time. In this study, we used the bubble chart to visualize the frequency of visitor comments, the emotional intensity of visitor comments, and the survey results of each welfare indicator. The visualization results provide a baseline against which future benchmark results can be evaluated. We suggest that giant pandas’ welfare literacy and conservation education programs be introduced at the Panda Base in the future as mechanism to track and influence and progress of these expected changes.

## 6. Conclusions

This study provides a first look into the giant panda welfare benchmark based on a co-design framework by scientists and the tourist community. The approach offers an overview of how visitors perceive, and conceive, giant panda welfare and sets a starting point for further educational and welfare literacy programs for academics and practitioners. Although visitor comments on animal welfare have been subjective, and often fragmented, the benchmark provides a framework to consolidate these individual opinions. At the same time, the benchmark is a collaborative outcome between animal welfare research (the Five Domain framework) and tourism studies, which ensures its theoretical alignment with current academic interests. The research-based framework establishes a basis from which to map and categorize visitors’ comments, with active engagement with comments helping to expand and contextualize panda welfare indicators. The co-design framework was thus created to encourage further collaboration between academics, practitioners, tourists, and animals as active agents in the experiential domain of tourism. Furthermore, our research follows other previous studies on engaged scholarship by indicating how research must have an applied context to it.

## 7. Limitations and Direction for Future Research

While the benchmark offers an overview of the current perception and experience of giant panda welfare in the tourist community, we acknowledge several limitations that need future exploration: First, qualitative analysis looking into the depth and meaning of visitor comments is mandatory to understand key gaps this study has demonstrated, for instance, the difference between the celebrity panda and regular pandas and the existing discrepancy between survey results and visitor comments. Also, qualitative analysis will help explain expressed emotions and sensations in the comments, which this study could not fully examine because of its focus on overseeing the general trend. Second, this study could benefit significantly from a longitudinal approach, which will establish further understanding of visitors’ perceptions and draw a baseline for promoting animal welfare in tourism in the future. 

## Figures and Tables

**Figure 1 animals-14-02137-f001:**
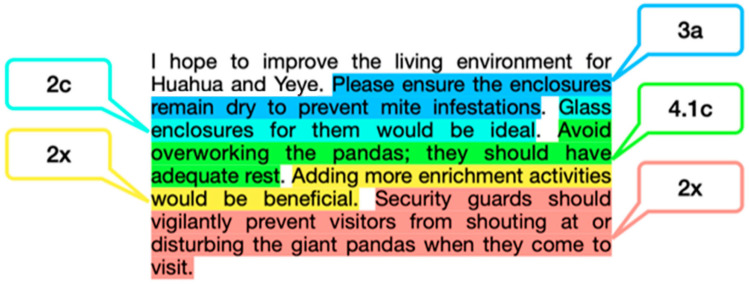
Coding a comment based on welfare indicators. Note. X means indicators emerged as the coding progressed. These indicators were not identified in Table 1, but were added into the table as indicators expanded.

**Figure 2 animals-14-02137-f002:**
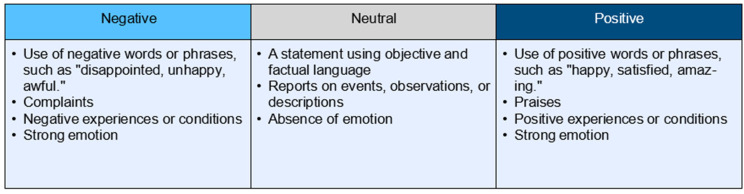
Sentiment categories for visitor comments.

**Figure 3 animals-14-02137-f003:**
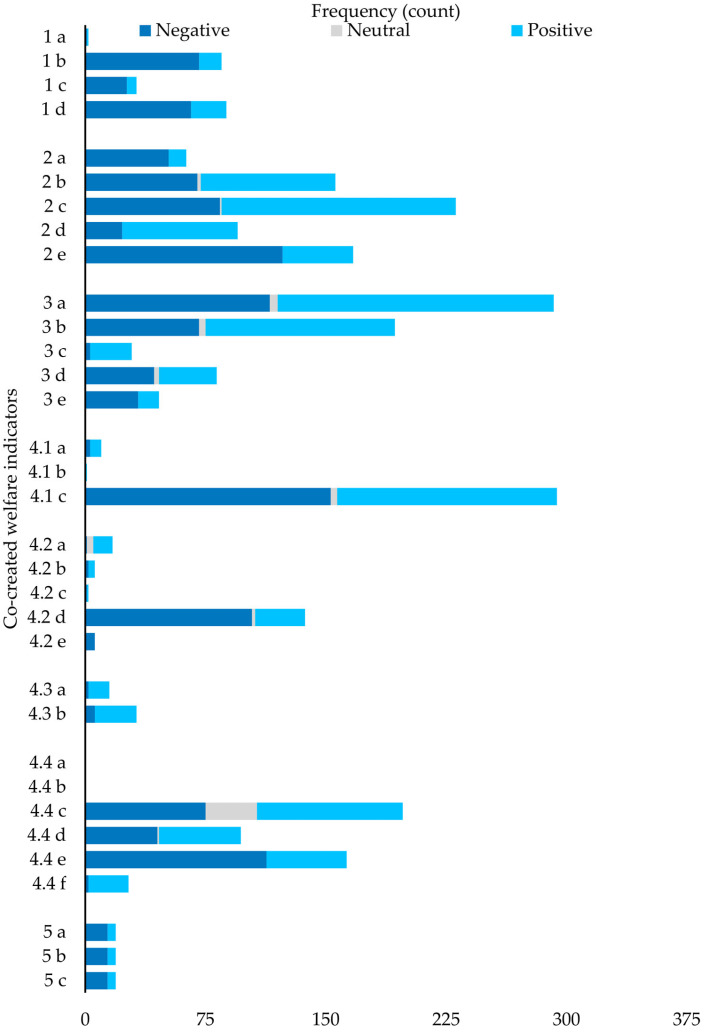
Visitor comments frequency and emotion distribution in the co-created framework.

**Figure 4 animals-14-02137-f004:**
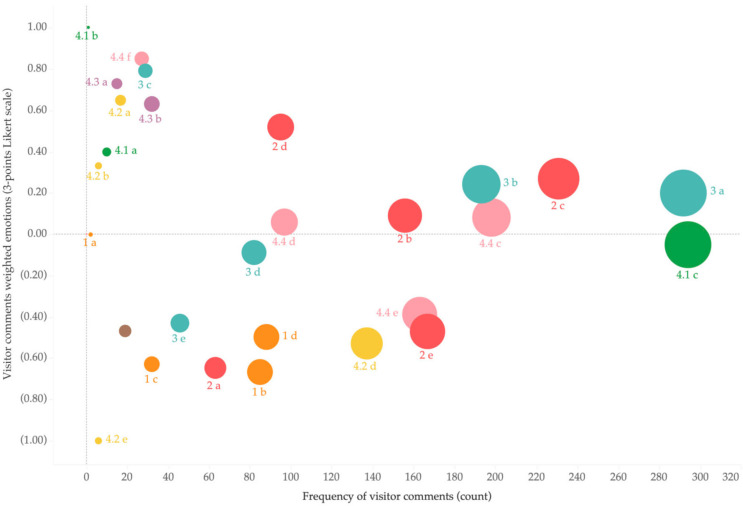
Emotion intensity and distribution of frequency in visitor comments.

**Figure 5 animals-14-02137-f005:**
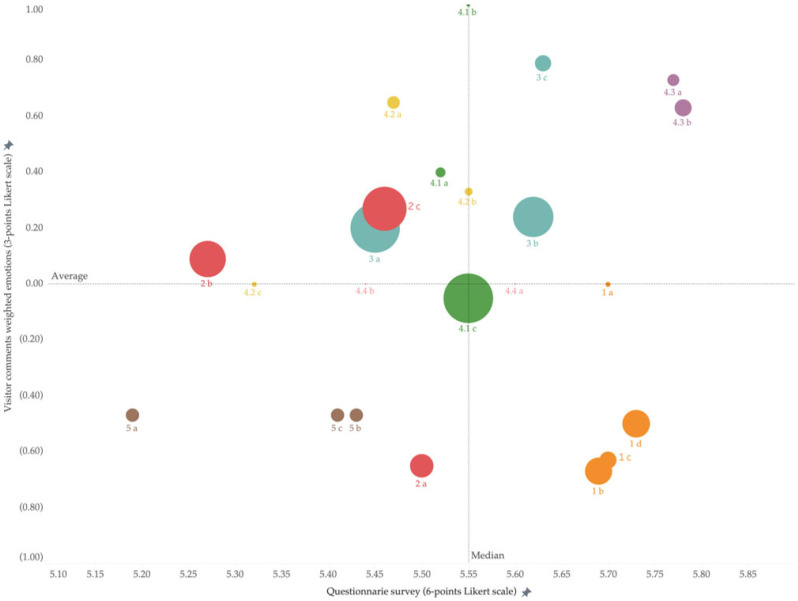
Comparison between comments, emotions detected in comments, and survey results.

**Figure 6 animals-14-02137-f006:**
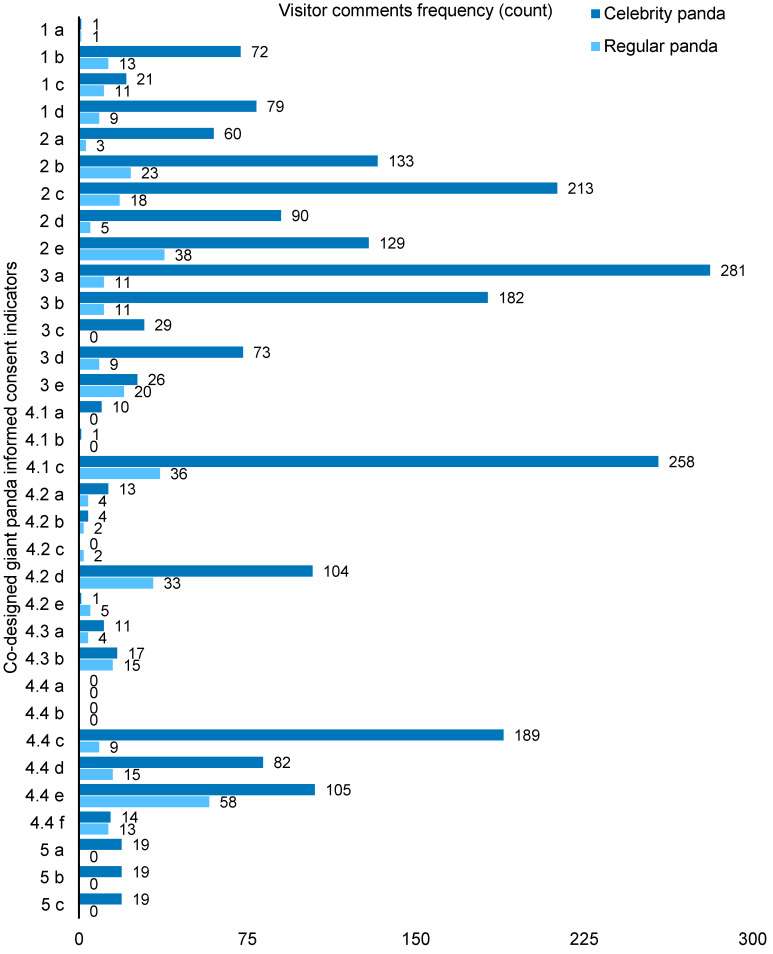
Visitor comments frequency for regular panda and celebrity pandas.

**Table 1 animals-14-02137-t001:** Giant panda welfare and consent indicators.

Indicator		Description	Survey Mean *
1. Nutrition		Water supply Food quantity Food quality Food variety	5.70 5.69 5.70 5.73
2. Physical environmental conditions		Spacious enclosures Enriched enclosures Enclosures protect pandas against thermal extremes	5.50 5.27 5.46
3. Health conditions		The panda does not suffer from acute or chronic injuries Healthcare treatment The panda is fit and attends to exercises	5.45 5.62 5.63
4. Behavioral interaction	4.1 Environment	Enclosures reflect the panda’s life stage Males and females are separated Multiple choices to move and be active	5.52 5.55 5.55
	4.2 Other animals	The panda socializes with a group The panda plays with its peers The panda enjoys community life	5.47 5.55 5.32
	4.3 Keepers	Keepers are kind and friendly Keepers are qualified and skilful	5.77 5.78
	4.4 Tourists	The panda is at ease despite the presence of tourists The panda can respond to my attention	5.60 5.44
5. Mental conditions		Pandas give consent to being used as tourist attractions. It is possible for pandass to express to humans their consent to being used as tourist attractions. Humans can recognize if pandas consent to being used as tourist attractions.	5.19 5.43 5.41

* The survey mean was calculated based on data used in Table 3 in Fennel and Guo [38].

**Table 2 animals-14-02137-t002:** Co-designed giant panda informed consent indicators.

Research-Based			Co-Designed		
Indicator		Description	Indicator		Description *
1. Nutrition		Water supply Food quantity Food quality Food variety	1. Nutrition		Water supply Food quantity Food quality Food variety
2. Physical environmental conditions		Spacious enclosures Enriched enclosures Enclosures protect pandas against thermal extremes	2. Physical environmental conditions		Spacious enclosures Enriched enclosures Enclosures protect pandas against thermal extremes Humidity Quiet environment
3. Health conditions		The panda does not suffer from acute or chronic injuries Healthcare treatment The panda is fit and attends to exercises	3. Health conditions		The panda does not suffer from acute or chronic injuries Healthcare treatment The panda is fit and attends to exercises Stereotypic behaviors Breeding
4. Behavioral interaction	4.1 Environment	Enclosures reflect the panda’s life stage Males and females are separated Multiple choices to move and be active	4. Behavioral interaction	4.1 Environment	Enclosures reflect the panda’s life stage Males and females are separated Multiple choices to move and be active
	4.2 Other animals	The panda socializes with a group The panda plays with its peers The panda enjoys community life		4.2 Other animals	The panda socializes with a group The panda plays with its peers The panda enjoys community life Giant pandas receive equal treatment Companion animals
	4.3 Keepers	Keepers are kind and friendly Keepers are qualified and skilful		4.3 Keepers	Keepers are kind and friendly Keepers are qualified and skilful
	4.4 Tourists	The panda is at ease despite the presence of tourists The panda can respond to my attention		4.4 Tourists	The panda is at ease despite the presence of tourists The panda can respond to my attention Information Accessibility Tourists empathize with pandas Tourists follow rules Panda-related activities
5. Mental conditions		Pandas give consent to being used as tourist attractions. It is possible for pandas to express to humans their consent to being used as tourist attractions. Humans can recognize if pandas consent to being used as tourist attractions.	5. Mental conditions		Pandas give consent to being used as tourist attractions. It is possible for pandas to express to humans their consent to being used as tourist attractions Humans can recognize if pandas are expressing their consent to being used as tourist attractions

* Colors used in this column: Red = Added by tourists; Blue = Not mentioned by tourists; Black = No change.

## Data Availability

Data is unavailable due to privacy or ethical restrictions.

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
