# Peer review of "Benchmarking Giant Panda Welfare in Tourism: A Co-Design Approach for Animals, Tourists, Managers, and Researchers"

_animals, 2024, doi:10.3390/ani14152137_

Round 1

Reviewer 1 Report

Comments and Suggestions for Authors

I commend the authors for this ms and hope that it can be published soon.

I only have some minor points, mostly formasl ones

- institutional affiliation is more than email address, please add your department etc (l 5 - 8)

- keywords: please do not repeat terms that are already in the title. Add scient. name, and one or two terms on methodology instead

- line 42: in referring to online please add date

- line 64 and 265: replace "escalating" by e.g. increasing, or any other non-negative term

- the citations in the text are inconsistent, e.g. in 152/153 why suddenly names instead of numbers, but when citing s.o. by name for clarity in the text (which is indeed helpful e.g.506/507) please ALSO add numbers

- fig 6 (line 410) is missing

- fg 4: please explain some more what and how it shows... 

Reviewer 2 Report

Comments and Suggestions for Authors

The literature gap, the aim of the paper, and the methodology are clearly stated in the submitted study. The sections on Results and Discussions are also well-prepared, and I have no objections to these parts of the paper. However, I have identified some minor corrections that I recommend:

In some instances, the authors write the number as 4839, and in others as 4,839. Please correct this throughout the text.

The text size in the Method section is inconsistent. Please revise this section for uniformity. The section is titled "Method," but does the study really use only one method? I recommend renaming this section to "Materials and Methods" or "Methodology."

The citation format throughout the text does not adhere to the journal’s instructions for authors. Please revise accordingly.

In section 3.3, the author mentions that 4839 comments were collected over 154 days, averaging 31 comments daily. This prompts a question: What is the monthly, daily, or weekly visitation rate of the Panda Base? Could you include data on the visitation rate in your study?

Regarding the Panda Base, I recommend that the authors add a map showing its location in China.

The same references are cited three times in the Discussion section, line 422. Please revise this for accuracy.

It is uncommon to cite references in the Conclusions section. If necessary, please move the citations from line 527 to the Discussion. Additionally, I recommend adding the limitations of the study to the Conclusion section. 
